**Data Availability Statement:** Data cannot be shared publicly beyond the Collaborative Individual

# Associations between symptoms of sleep-disordered breathing and maternal sleep patterns with late stillbirth: Findings from an individual participant data meta-analysis

**Robin S. Cronin**[1]*, **Jessica Wilson**[1], **Adrienne Gordon**[2], **Minglan Li**[1], **Vicki M. Culling**[1], **Camille H. Raynes-Greenow**[3], **Alexander E. P. Heazell**[4], **Tomasina Stacey**[5], **Lisa M. Askie**[6], **Edwin A. Mitchell**[1], **John M. D. Thompson**[1], **Lesley M. E. McCowan**[1], **Louise M. O'Brien**[7]

1 Departments of Obstetrics and Gynaecology, and Paediatrics: Child and Youth Health, Faculty of Medical and Health Sciences, University of Auckland, Auckland, New Zealand, 2 Discipline of Obstetrics, Gynaecology and Neonatology, University of Sydney, Sydney, Australia, 3 Sydney School of Public Health, University of Sydney, Sydney, Australia, 4 Division of Developmental Biology & Medicine, Maternal and Fetal Health Research Centre, School of Medical Sciences, University of Manchester, Manchester, England, United Kingdom, 5 Department of Nursing and Midwifery, School of Human and Health Sciences, University of Huddersfield, Huddersfield, England, United Kingdom, 6 National Health and Medical Research Council Clinical Trials Centre, University of Sydney, Sydney, Australia, 7 Departments of Neurology Sleep Disorders Center, and Obstetrics and Gynecology, University of Michigan, Ann Arbor, Michigan, United States of America

* r.cronin@auckland.ac.nz, robincronin@xtra.co.nz

## Abstract

### Background and objectives

Sleep-disordered breathing (SDB) affects up to one third of women during late pregnancy and is associated with adverse pregnancy outcomes, including hypertension, diabetes, impaired fetal growth, and preterm birth. However, it is unclear if SDB is associated with late stillbirth (≥28 weeks' gestation). The aim of this study was to investigate the relationship between self-reported symptoms of SDB and late stillbirth.

### Methods

Data were obtained from five case-control studies (cases 851, controls 2257) from New Zealand (2 studies), Australia, the United Kingdom, and an international study. This was a secondary analysis of an individual participant data meta-analysis that investigated maternal going-to-sleep position and late stillbirth, with a one-stage approach stratified by study and site. Inclusion criteria: singleton, non-anomalous pregnancy, ≥28 weeks' gestation. Sleep data ('any' snoring, habitual snoring ≥3 nights per week, the Berlin Questionnaire [BQ], sleep quality, sleep duration, restless sleep, daytime sleepiness, and daytime naps) were collected by self-report for the month before stillbirth. Multivariable analysis adjusted for known major risk factors for stillbirth, including maternal age, body mass index (BMI kg/m2), ethnicity, parity, education, marital status, pre-existing hypertension and diabetes, smoking, recreational drug use, baby birthweight centile, fetal movement, supine going-to-

Participant Data Meta-analysis of Sleep and Stillbirth (CRIBSS) group as no individual participating study obtained consent from participants to make the data publically available. Furthermore, because stillbirth is uncommon there is potential for participants to be identifiable. Contact information for the CRIBSS Data Access Committee is The CRIBSS Data Centre, Department of Obstetrics and Gynaecology, Faculty of Medical and Health Sciences, University of Auckland, Private Bag 92019, Auckland Mail Centre, Auckland 1142.

**Funding:** Mrs Cronin reports grants from Health Research Council of New Zealand (12/372); Cure Kids (5357); Mercia Barnes Trust; Nurture Foundation; University of Auckland Faculty Research Development Fund (3700696)., grants from 2016 TransTasman Red Nose/Curekids (6601), grants from Sir John Logan Campbell Medical Trust, during the conduct of the study; Ms. Wilson reports grants from 2016 TransTasman Red Nose/Curekids (6601); Dr. Gordon reports grants from Stillbirth Foundation Australia, during the conduct of the study; other from NHMRC Early Career Fellowship #1089898, outside the submitted work; Dr. Li reports grants from Health Research Council of New Zealand (12/372); Cure Kids (5357); Mercia Barnes Trust; Nurture Foundation; University of Auckland Faculty Research Development Fund (3700696)., grants from 2016 TransTasman Red Nose/Curekids (6601), during the conduct of the study; Dr. Culling has nothing to disclose; Associate Professor Raynes-Greenow reports grants from Stillbirth Foundation Australia, during the conduct of the study; other from NHMRC Career Development Fellowship #1087062, outside the submitted work; Professor Heazell reports grants from Action Medical Research, during the conduct of the study; grants from Tommy's, grants from NIHR, outside the submitted work; Dr. Stacey reports grants from Cure Kids (3537), Nurture Foundation, Auckland District Health Board Charitable Trust., grants from Health Research Council of New Zealand (12/372); Cure Kids (5357); Mercia Barnes Trust; Nurture Foundation; University of Auckland Faculty Research Development Fund (3700696)., during the conduct of the study; Professor Askie has nothing to disclose; Professor Mitchell reports grants from Cure Kids (Grant 3537), Nurture Foundation, Auckland District Health Board Trust., grants from Health Research Council of New Zealand (12/372); Cure Kids (5357); Mercia Barnes Trust; Nurture Foundation; University of Auckland Faculty Research Development Fund (3700696)., grants from 2016 TransTasman Red Nose/Curekids (6601), during the conduct of the study;

sleep position, getting up to use the toilet, measures of SDB and maternal sleep patterns significant in univariable analysis (habitual snoring, the BQ, sleep duration, restless sleep, and daytime naps). Registration number: PROSPERO, CRD42017047703.

## Results

In the last month, a positive BQ (adjusted odds ratio [aOR] 1.44, 95% confidence interval [CI] 1.02–2.04), sleep duration >9 hours (aOR 1.82, 95% CI 1.14–2.90), daily daytime naps (aOR 1.52, 95% CI 1.02–2.28) and restless sleep greater than average (aOR 0.62, 95% CI 0.44–0.88) were independently related to the odds of late stillbirth. 'Any' snoring, habitual snoring, sleep quality, daytime sleepiness, and a positive BQ excluding the BMI criterion, were not associated.

## Conclusion

A positive BQ, long sleep duration >9 hours, and daily daytime naps last month were associated with increased odds of late stillbirth, while sleep that is more restless than average was associated with reduced odds. Pregnant women may be reassured that the commonly reported restless sleep of late pregnancy may be physiological and associated with a reduced risk of late stillbirth.

## Introduction

The loss of a baby from stillbirth has detrimental consequences for the family and the community [1]. The causes of many stillbirths are unexplained [2, 3]. Sleep-disordered breathing (SDB), ranging from snoring to obstructive sleep apnoea (OSA), is common during pregnancy. The cardinal symptom, habitual snoring ≥3 nights per week, affects up to 35% of women in the third trimester [4, 5], and up to 85% of women with pre-eclampsia [6], while objective measures of OSA are estimated to affect between 8% and 26% of pregnant women [7, 8]. SDB is a risk factor for adverse pregnancy outcomes, including gestational hypertension and pre-eclampsia [4, 9, 10], hyperglycaemia [11–13], impaired fetal growth [14–19], and early-term and/or preterm birth [9, 14, 20–22]. SDB is exacerbated by obesity, advanced gestation, and the supine sleep position [23], all of which are themselves associated with an increased risk of late stillbirth [24]. Therefore, pregnant women with SDB may have an increased risk of late stillbirth (≥28 weeks' gestation) and this risk may be magnified if women settle to sleep supine, however the data is lacking.

Importantly, the association between SDB and maternal sleep patterns (sleep quality, sleep duration, restless sleep, daytime sleepiness, and daytime naps) with late stillbirth is inconsistent across studies. A meta-analysis [25], which included the comparison of stillbirth in women with and without SDB as an outcome measure, using subjective (self-reported snoring) [26, 27] and objective (OSA) [9, 28, 29] measurements, reported no association between SDB and stillbirth. The relationship between sleep duration and late stillbirth was reported in several case-control [26, 30–32] and cross sectional studies [33], however, the results are not consistent in identifying an association. Subjective sleep quality was also not associated with stillbirth in a cross-sectional [33] and case-control [32] study. Other case-control studies [26, 30] reported that daily naps, compared to no naps, were independently associated with late stillbirth. These inconsistencies may be due to differing measurements of these aspects of

other from Cure Kids, outside the submitted work; Associate Professor Thompson reports grants from Cure Kids (3537), Nurture Foundation, Auckland District Health Board Charitable Trust., grants from Health Research Council of New Zealand (12/372); Cure Kids (5357); Mercia Barnes Trust; Nurture Foundation; University of Auckland Faculty Research Development Fund (3700696)., grants from 2016 TransTasman Red Nose/ Curekids (6601), during the conduct of the study; Professor McCowan reports grants from Cure Kids (537), Nurture Foundation, Auckland District Health Board Trust., grants from Health Research Council of New Zealand (12/372); Cure Kids (5357); Mercia Barnes Trust; Nurture Foundation; University of Auckland Faculty Research Development Fund (3700696)., grants from 2016 TransTasman Red Nose/Curekids (6601), during the conduct of the study. Associate Professor O'Brien reports grants from American Sleep Medicine Foundation, grants from ResMed, outside the submitted work.

**Competing interests:** Associate Professor O'Brien reports grants from ResMed outside the submitted work. This does not alter our adherence to PLOS ONE policies on sharing data and materials.

maternal sleep between studies, or because some studies did not adjust for potential confounders (such as maternal body mass index [BMI kg/m2] and maternal age). Furthermore, as late stillbirth is a relatively rare event, ranging from 1·3 to 8·8/1000 births in high-income countries [3], individual studies have been underpowered to investigate interactions between supine going-to-sleep position and late stillbirth in women with SDB compared to those without.

The triple risk model [34] suggests that late stillbirth may be the culmination of an interplay between stressors (e.g. SDB, supine going-to-sleep position), maternal risk factors (e.g. obesity, age), and fetal-placental vulnerability (e.g. impaired fetal growth, placental dysfunction). Exploration of possible biological pathways [35] of the association of adverse pregnancy outcomes related to SDB suggests that there are multifactorial mechanisms, including sympathetic activation, oxidative stress, inflammation, and endothelial dysfunction, which contribute to maternal cardiovascular dysfunction, metabolic derangement, placental dysfunction, and fetal compromise. Thus, it is plausible that when a mother is in the supine position in late pregnancy and there is reduced maternal-fetal blood flow from aortocaval compression [36, 37], the addition of partial airway collapse with SDB may exacerbate fetal compromise in a vulnerable fetus.

Since SDB and maternal sleep patterns are potentially modifiable during pregnancy (such as lateral position for supine-dependent snoring, continuous positive airway pressure for OSA, and frequency of daytime naps), it is possible that screening and management of these aspects of maternal sleep during pregnancy may support reduction in the rate of late stillbirth. However, there is a need to assess the current evidence from individual studies that have collected data on maternal sleep and stillbirth to determine if they are associated with late stillbirth.

We established the Collaborative Individual Participant Data (IPD) Meta-analysis of Sleep and Stillbirth (CRIBSS) group to address if maternal going-to-sleep position was associated with late stillbirth. This included pre-specified secondary questions on symptoms of SDB and maternal sleep patterns [38], including 1) is SDB associated with late stillbirth, and 2) is supine going-to-sleep position associated with greater risk of late stillbirth in women with SDB compared to those without?

## Materials and methods

The study population comprised cases with late stillbirth and controls with ongoing pregnancies from the CRIBSS data. This IPD meta-analysis was registered with the PROSPERO register of systematic reviews (CRD42017047703) and followed the IPD meta-analysis protocol [38], search strategy [24], risk of bias for non-randomised studies (ROBINS-E tool) [39], and published results [24]. Five international case-control studies [26, 27, 30–32] that collected maternal going-to-sleep position and late stillbirth data were included in this pooled IPD meta-analysis.

Participant level inclusion criteria were singleton, non-anomalous pregnancy, ≥28 weeks' gestation. Exclusion criteria were multiple pregnancy, major congenital abnormality, gestation <28 weeks' when pregnancy sleep data was collected, termination of pregnancy at ≥28 weeks', and receiving an intervention that may have affected going-to-sleep position. Maternal sleep data were collected by self-report via face-to-face interview [26, 27, 30, 31] or online survey [32] within six weeks after stillbirth in cases or at a matched gestation in controls.

Late stillbirth, using the international definition of stillbirth [40], "a baby born with no signs of life at or after 28 weeks' gestation," was the primary outcome. The analysis included intrapartum stillbirth, with the rationale that the exact time of the stillbirth may be uncertain and that SDB may result in a vulnerable baby that is unable to tolerate labour.

## Data analysis

This was a prespecified secondary analysis of an IPD meta-analysis that investigated maternal going-to-sleep position and late stillbirth, with a one-stage approach stratified by study and site. A detailed statistical analysis plan, prior to the analysis, has been published.[25] Prespecified potential covariates were: maternal age, earliest pregnancy BMI, ethnicity, parity, education level, marital status, pre-existing hypertension or diabetes, smoking, recreational drug use, supine going-to-sleep position, fetal movements, infant birthweight by customised centiles, and measures of SDB and sleep patterns ('any' snoring, habitual snoring, the Berlin Questionnaire [BQ], Epworth Sleepiness Scale [ESS], sleep quality, sleep restlessness, and sleep duration). Frequency of getting up to use the toilet and daytime naps were also included as these are previously reported [26, 30–32] independent risk factors for late stillbirth. Where data exists for multiple time frames, only data for the month prior to the stillbirth were used in the analysis. In cases where the last month data were not available, data collected for the 'last week' [31] were used.

There are currently no validated tools for SDB screening during pregnancy, therefore we investigated habitual snoring, a positive BQ [41], and daytime sleepiness using the ESS [42] as proxy indicators. The BQ [41] was developed to identify individuals at risk of OSA in non-pregnant primary care populations and has three categories 1) snoring frequency, loudness, and witnessed apnoea, 2) daytime sleepiness, and 3) BMI >30 and hypertension, with a positive BQ requiring two positive categories. The ESS [42] is a subjective measure of daytime sleepiness with eight questions about the likelihood of dozing off in specified situations, ranging from unlikely (in a car stopped for a few minutes in traffic) to highly likely (lying down to rest in the afternoon). The ESS is coded as 0 = never doze, 1 = slight chance, 2 = moderate chance, and 3 = high chance, with a positive ESS screen indicating clinical levels of daytime sleepiness defined as $\geq$10.

Data on the usual duration of overnight sleep were also collected. The reference for sleep duration was defined as 6 to 9 hours, with duration categorised as <6, 6–9, or >9 hours. Restless sleep and sleep quality were each single questions, with 'average' restlessness and 'average' sleep quality as the reference group.

A one-stage approach to meta-analysis was used, so that the data from the participating eligible studies (Table 1) were included in a single model. Logistic regression models were used for the binary outcome. A fixed study effect and study site effect were included in the model specification as strata. Univariable analysis was performed to evaluate the association between the measures of SDB and maternal sleep patterns and the odds of late stillbirth. A multivariable model was developed incorporating prespecified covariates [38] available in all the studies (Appendix 1 in the S1 Protocol) and measures of SDB and maternal sleep patterns that were significant in univariable analysis (Table 1). Some covariates (habitual snoring, the BQ, sleep quality, restless sleep, daytime naps, daytime sleepiness using the ESS, and getting up to use the toilet) were not available in all participating studies (S1 Fig).

The interaction between supine going-to-sleep position and common measures of SDB (habitual snoring and the BQ) and sleep duration were assessed in bi-variable regression models. Significant interactions were then added to the multivariable model as described above. Estimates of the risk of late stillbirth were reported as odds ratio (OR) with 95% confidence intervals (95% CI). For missing data in each individual study, imputation was not undertaken. Statistical analyses were performed using SAS, version 9.4 (SAS Institute Inc., Cary NC USA).

Each individual study obtained ethical approval [26, 27, 30–32]. Approval for the IPD meta-analysis was obtained from the New Zealand Health and Disability Ethics Committee (NTX/06/05/054/AM06).

**Table 1. Study level characteristics and measured sleep-related factors in participating studies.**

| Study level characteristics | The Auckland Stillbirth Study | Sydney Stillbirth Study | New Zealand Multicentre Stillbirth Study | Midlands and North of England Stillbirth Study | Study of Trends and Associated Risks for Stillbirth Study |
|---|---|---|---|---|---|
| | Stacey et al (2011)[6] | Gordon et al (2015)[9] | McCowan et al (2017)[4] | Heazell et al (2017)[8] | O'Brien et al (2018)[7] |
| Location | Auckland, New Zealand | Sydney, Australia | New Zealand | United Kingdom | International |
| Years of recruitment | July 2006 to June 2009 | January 2006 to December 2011 | February 2012 to December 2015 | April 2014 to March 2016 | September 2012 to August 2014 |
| Study design | Prospective population-based case-control | Prospective population-based case-control | Prospective population-based case-control | Prospective population-based case-control | Nested case-control with uncontrolled cohort |
| Population | Non-anomalous singleton pregnancy, ≥28 weeks' gestation, from three health regions in Auckland, New Zealand | Non-anomalous singleton pregnancy, ≥32 weeks' gestation, from nine tertiary maternity facilities in metropolitan Sydney, Australia | Non-anomolous singleton pregnancy, ≥28 weeks' gestation, from seven health regions throughout New Zealand | Non-anomalous singleton pregnancy, ≥28 weeks' gestation, from 41 maternity facilities in the United Kingdom | Singleton pregnancy, ≥28 weeks' gestation, fluent in English, from 16 high, middle, and low income countries |
| Stated main outcome measure | Maternal snoring, daytime sleepiness, and sleep position at the time of going to sleep and on waking (left side, right side, back, and other) | Risk factors for late-pregnancy stillbirth with a particular focus on those risks that are potentially modifiable | The adjusted odds of late stillbirth associated with self-reported going-to-sleep position, on the last night | Maternal sleep practices pregnancy | To investigate, in an international cohort, whether maternal sleep practices are related to late stillbirth |
| Measured sleep-related factors | Sleep position (going-to-sleep, waking) | Sleep position | Sleep position (going-to-sleep, waking) | Sleep position (going-to-sleep, waking) | Sleep position (going-to-sleep, waking) |
| | Snoring presence | | Snoring presence | Snoring presence | Snoring presence |
| | Sleep duration | Snoring presence | Sleep duration | Sleep duration | Sleep duration |
| | Sleep quality | Sleep duration | Sleep quality | Sleep quality | Sleep quality |
| | Sleep restlessness | Sleep quality | Sleep restlessness | Sleep restlessness | Sleep restlessness |
| | Getting up to toilet | Sleep restlessness | Getting up to toilet | Getting up to toilet | Getting up to toilet |
| | Daytime naps | Getting up to toilet | Daytime naps | Daytime naps | Daytime naps |
| | Epworth Sleepiness Scale | Daytime naps | Epworth Sleepiness Scale | Epworth Sleepiness Scale | Epworth Sleepiness Scale |
| | Sleep apnoea | Epworth Sleepiness Scale | Berlin Questionnaire | Berlin Questionnaire | Berlin Questionnaire |
| | Night waking | Berlin Questionnaire | Night waking | - | Night waking |
| | - | - | Restless legs | Restless legs | Restless legs |
| | - | - | Sleep latency | Sleep latency | Sleep latency |
| | - | - | Position changes | Position changes | Position changes |
| | - | - | Insomnia | Insomnia | Insomnia |
| | - | - | Bed size and side | Bed size and side | Bed size and side |
| | - | - | Pillow(s) placement | Pillow(s) placement | Pillow(s) placement |
| | - | - | Sleep partners | Sleep partners | Sleep partners |
| | - | - | Sleep advice | Sleep advice | - |
| | - | - | - | Sleep medication | Sleep medication |
| | - | - | Sleep chronotype | - | - |
| Time frames of measured sleep factors | Pre-pregnancy | Pre-pregnancy | - | Pre-pregnancy | Pre-pregnancy |
| | During pregnancy | During pregnancy | During pregnancy | During pregnancy | During pregnancy |
| | Last month | Last month | - | Last 4 weeks | Last 4 weeks |
| | - | Last two weeks | Last week | - | - |
| | - | - | - | Last week | Last week |
| | Last night | - | Last night | Last night | Last night |
| Data collection | Interview and clinical records | Interview and clinical records | Interview and clinical records | Interview and clinical records | Online survey |

## Results

Participants comprised 851 late stillbirth cases and 2257 controls with ongoing pregnancies from five eligible case-control studies (Fig 1): the Auckland Stillbirth Study [26], the New

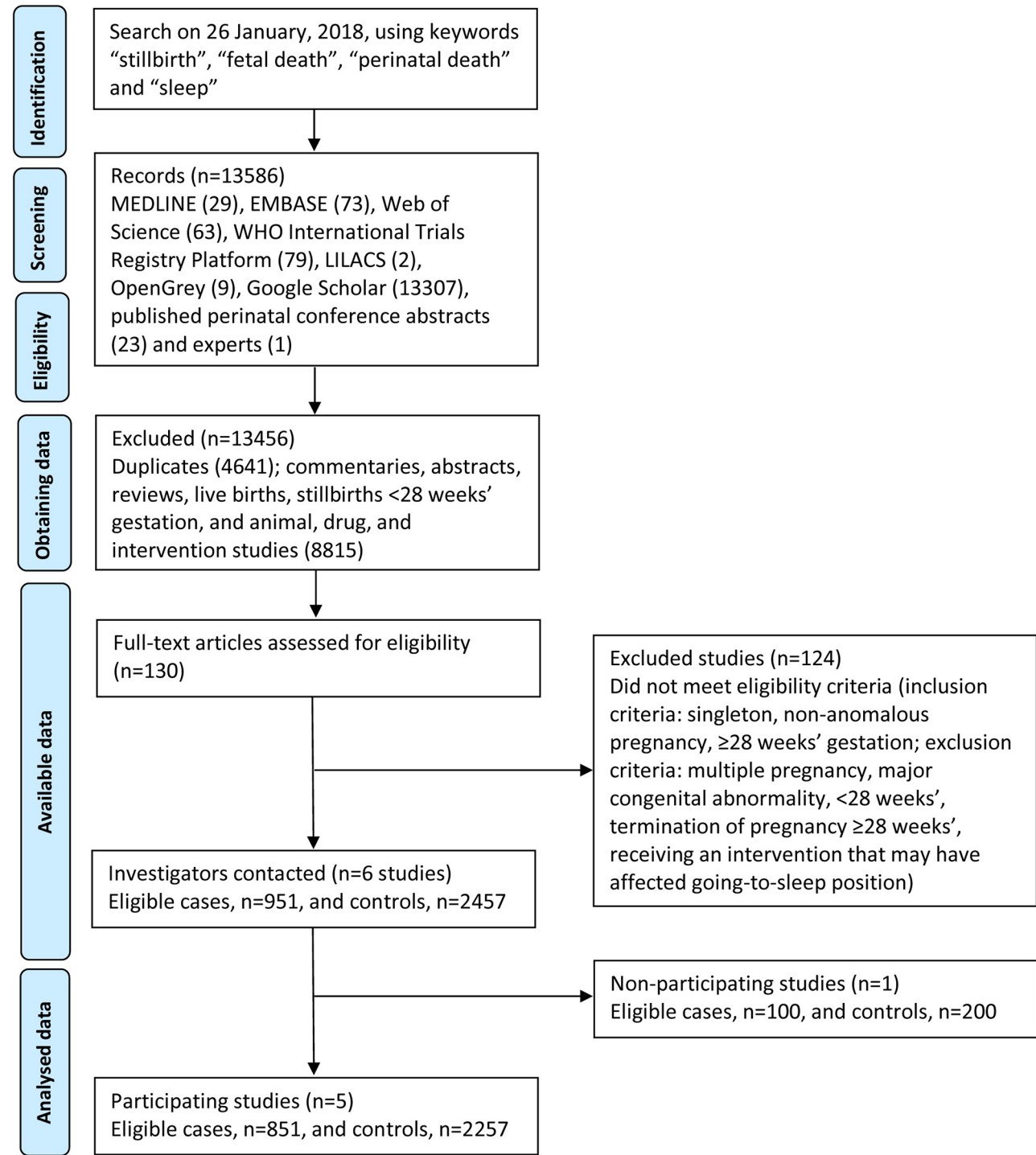

**Fig 1. PRISMA study population flow chart.** Adapted from EClinicalMedicine, Vol 10, Authors: Cronin, RS., Li, M., Thompson, JMD., Gordon, A., Raynes-Greenow, CH., Heazell, AEP., Stacey, T., Culling, VM., Bowring, V., Anderson, NH., O'Brien, LM., Mitchell, EA., Askie, LM., McCowan, LME, An Individual Participant Data Meta-analysis of Maternal Going-to-Sleep Position, Interactions with Fetal Vulnerability, and the Risk of Late Stillbirth, Pages 49–57., Copyright (2019), with permission from Elsevier.

Zealand Multicentre Stillbirth Study [31], the Sydney Stillbirth Study [27], the UK Midlands and North of England Stillbirth Study [30], and the International Study of Trends and Associated Risks for Stillbirth Study [32], comprising women of many ethnicities [24].

Differences in maternal and pregnancy characteristics, infant size, and going-to-sleep position between cases and controls have been previously reported (S1 Table) [24]. 'Any' snoring (cases n = 473, 56.0%; controls, n = 1182, 54.1%), sleep quality (fairly bad to very bad, cases n = 248, 33.5%; controls, n = 703, 35.3%), daytime sleepiness (positive ESS score ≥10, cases n = 128, 17.5%; controls n = 312, 15.8%), and frequency of getting up to use the toilet (≥1 per night, cases n = 667, 90.0%; controls, n = 1820, 91.5%) last month were not associated with late stillbirth in the univariable analysis.

Long sleep duration >9 hours last month (cases n = 78, 10.5%; controls, n = 129, 6.5%) was independently associated with late stillbirth compared to sleep duration of 6 to 9 hours (adjusted odds ratio [aOR] 1.82, 95% CI 1.14–2.90) (Table 2). Reporting a daily daytime nap last month (cases n = 139, 23.7%; controls, n = 216, 12.8%) compared to never reporting a daytime nap was associated with an increase in the odds of late stillbirth (aOR 1.52, 95% CI 1.02–2.28). In addition, a positive BQ (cases n = 176, 30.0%; controls, n = 370, 21.8%) was associated with late stillbirth (aOR 1.44, 95% CI 1.02–2.04), however, when BMI >30 was removed from the BQ score, a positive BQ showed no significant association with stillbirth (aOR 0.81, 95% CI 0.54–1.21). Restless sleep greater than average last month (cases n = 225, 38.3%; controls, n = 761, 45.2%) was associated with a reduction in the odds of late stillbirth (aOR 0.62, 95% CI 0.44–0.88).

Women who had a stillbirth, 689 cases from four participating studies [27, 30–32], were asked what time of day they thought their baby had died: 34.8% (n = 240, or 52.3% of 459 cases who could recall a time of day) reported that they thought their baby had died overnight, 19.4% (n = 134) reported afternoon-evening, 11.8% (n = 81) morning, 0.6% (n = 4) during a daytime nap, and 33.4% (n = 230) were unsure (Fig 2).

Interactions were assessed between supine going-to-sleep position and habitual snoring, a positive BQ including BMI, sleep duration >9 hours, and restless sleep greater than average last month (Table 3). Interactions for a positive BQ (p = 0.56), sleep duration >9 hours (p = 0.99), and restless sleep greater than average (p = 0.98) were not statistically significant. There was a significant interaction between habitual snoring and supine going-to-sleep position (multivariable interaction p value = 0.001). The combined effect of supine going-to-sleep position and habitual snoring resulted in a reduced odds of late stillbirth in the multivariable model than would be expected. (Table 3).

## Discussion

### Main findings

Our study has demonstrated that a positive BQ, long sleep duration >9 hours, and a daily daytime nap in the last month, were each associated with increased odds of late stillbirth. In contrast, restless sleep greater than average in the last month was protective for late stillbirth. The associations between these aspects of maternal sleep and late stillbirth were adjusted for pre-specified covariates [38] available in all the studies (S 1), and measures of SDB and maternal sleep patterns significant in univariable analysis (Table 1).

The ~50% prevalence of 'any' snoring and habitual snoring ≥3 nights per week between 17–24% was within the range reported in the pregnancy literature [4, 43–46]. 'Any' snoring, habitual snoring, sleep quality, and daytime sleepiness using the ESS, was not associated with late stillbirth (Table 2). This is consistent with previous studies: snoring [26, 27, 33], sleep quality [33], and daytime sleepiness [47].

**Table 2. Subjective indicators of sleep-disordered breathing and maternal sleep patterns in participating case-control studies and pooled IPD meta-analysis.**

| Characteristic | TASS Stacey et al (2011) [26] Case | Control | SSS Gordon et al (2015) [27] Case | Control | MCSS McCowan et al (2017)[31] Case | Control | MiNESS Heazell et al (2017) [30] Case | Control | STARS O'Brien et al (2018) [32] Case | Control | Collaborative Individual Participant Data of Going-to-sleep and Stillbirth (CRIBSS) analysis Case | Control | Univariable odds ratio (95% CI) | Adjusted odds ratio (95% CI) |
|---|---|---|---|---|---|---|---|---|---|---|---|---|---|---|
| **Total participants** | 155 (33.8) | 304 (66.2) | 103 (34.9) | 192 (65.1) | 163 (22.5) | 560 (77.5) | 288 (28.2) | 733 (71.8) | 142 (23.3) | 468 (76.7) | 851 (27.4) | 2257 (72.6) | | |
| **Going-to-sleep position (last two weeks)** | | | | | | | | | | | | | | |
| Non-supine | 104 (87.4) | 242 (94.5) | 84 (89.4) | 183 (97.9) | 139 (88.0) | 539 (96.4) | 254 (93.0) | 698 (96.7) | 124 (96.9) | 355 (97.0) | 705 (91.3) | 2017 (96.5) | 1 | 1 |
| Supine | 15 (12.6) | 14 (4.5) | 10 (10.6) | 4 (2.1) | 19 (12.0) | 20 (3.6) | 19 (7.0) | 24 (3.3) | 4 (3.1) | 11 (3.0) | 67 (8.7) | 73 (3.5) | **2.85 (2.01–4.05)** | **3.06 (1.77–5.28)** |
| **Snoring 'any' (during pregnancy)** | | | | | | | | | | | | | | |
| No | 86 (55.5) | 175 (57.6) | 49 (47.6) | 93 (48.4) | 59 (36.2) | 255 (45.5) | 118 (41.0) | 300 (41.2) | 59 (43.7) | 179 (44.8) | 371 (44.0) | 1002 (45.9) | 1 | - |
| Yes | 69 (44.5) | 129 (42.4) | 54 (52.4) | 99 (51.6) | 104 (63.8) | 305 (54.5) | 170 (59.0) | 428 (58.8) | 76 (56.3) | 221 (55.2) | 473 (56.0) | 1182 (54.1) | 1.11 (0.95–1.31) | - |
| **Habitual snoring ≥ 3 nights/week (last month)** | | | | | | | | | | | | | | |
| No | - | - | - | - | 129 (79.1) | 494 (88.2) | 192 (74.7) | 544 (81.1) | 95 (76.6) | 286 (76.5) | 416 (76.5) | 1324 (82.5) | 1 | 1 |
| Yes | - | - | - | - | 34 (20.9) | 66 (11.8) | 65 (25.3) | 127 (18.9) | 29 (23.4) | 88 (23.5) | 128 (23.5) | 281 (17.5) | **1.40 (1.10–1.78)** | 1.04 (0.74–1.47) |
| **Berlin Questionnaire** | | | | | | | | | | | | | | |
| Negative screen | - | - | - | - | 106 (65.0) | 463 (82.7) | 195 (67.7) | 534 (72.9) | 110 (80.9) | 331 (81.7) | 411 (70.0) | 1328 (78.2) | 1 | 1 |
| Positive screen | - | - | - | - | 57 (35.0) | 97 (17.3) | 93 (32.3) | 199 (27.2) | 26 (19.1) | 74 (18.3) | 176 (30.0) | 370 (21.8) | **1.52 (1.22–1.89)** | **1.44 (1.02–2.04)** |
| **Restless sleep (last month)** | | | | | | | | | | | | | | |
| Less than average | - | - | - | - | 73 (44.8) | 276 (49.3) | 109 (37.8) | 214 (29.3) | 47 (34.6) | 95 (24.2) | 229 (39.0) | 585 (34.7) | 1.00 (0.77–1.28) | 1.08 (0.78–1.50) |
| Average | - | - | - | - | 41 (25.1) | 127 (22.7) | 61 (21.2) | 110 (15.0) | 31 (22.8) | 101 (25.7) | 133 (22.7) | 338 (20.1) | 1 | 1 |
| Greater than average | - | - | - | - | 49 (30.1) | 157 (28.0) | 118 (41.0) | 407 (55.7) | 58 (42.6) | 197 (54.1) | 225 (38.3) | 761 (45.2) | **0.75 (0.59–0.97)** | **0.62 (0.44–0.88)** |
| **Sleep duration overnight (last month)** | | | | | | | | | | | | | | |
| <6 hours | 30 (19.4) | 45 (14.8) | - | - | 27 (16.5) | 79 (14.1) | 78 (27.1) | 212 (29.1) | 7 (5.1) | 46 (11.4) | 142 (19.1) | 382 (19.1) | 1.06 (0.85–1.33) | 0.77 (0.55–1.07) |
| 6–9 hours | 104 (67.1) | 233 (76.6) | - | - | 123 (75.5) | 452 (80.7) | 179 (62.1) | 477 (65.5) | 116 (85.3) | 321 (79.9) | 522 (70.4) | 1483 (74.4) | 1 | 1 |
| >9 hours | 21 (13.5) | 26 (8.6) | - | - | 13 (8.0) | 29 (5.2) | 31 (10.8) | 39 (5.4) | 13 (9.6) | 35 (8.7) | 78 (10.5) | 129 (6.5) | **1.67 (1.23–2.26)** | **1.82 (1.14–2.90)** |
| **Daytime naps (last month)** | | | | | | | | | | | | | | |
| Never | - | - | - | - | 33 (26.4) | 109 (28.0) | 56 (44.8) | 157 (40.4) | 36 (28.8) | 123 (31.6) | 125 (21.3) | 389 (23.1) | 1 | 1 |
| Occasionally | - | - | - | - | 63 (32.1) | 248 (36.1) | 96 (49.0) | 333 (48.4) | 37 (18.9) | 107 (15.6) | 196 (33.4) | 688 (40.8) | 0.90 (0.69–1.17) | 0.92 (0.66–1.30) |
| Often | - | - | - | - | 28 (22.1) | 133 (33.8) | 66 (52.0) | 149 (37.8) | 33 (26.0) | 112 (28.4) | 127 (21.6) | 394 (23.4) | 1.03 (0.77–1.38) | 0.91 (0.62–1.33) |
| Everyday | - | - | - | - | 39 (28.1) | 70 (32.4) | 70 (50.4) | 93 (43.1) | 30 (21.6) | 53 (24.5) | 139 (23.7) | 216 (12.8) | **2.06 (1.52–2.78)** | **1.52 (1.02–2.28)** |
| **Daytime sleepiness screen (Epworth Sleepiness Scale) (last month)** | | | | | | | | | | | | | | |
| Negative <10 | 136 (87.7) | 270 (88.8) | - | - | 129 (79.0) | 479 (85.5) | 244 (85.3) | 612 (84.0) | 96 (74.4) | 302 (79.0) | 604 (82.5) | 1660 (84.2) | 1 | - |

(*Continued*)

**Table 2.** (Continued)

| Characteristic | TASS Stacey et al (2011) [26] | | SSS Gordon et al (2015) [27] | | MCSS McCowan et al (2017)[31] | | MiNESS Heazell et al (2017) [30] | | STARS O'Brien et al (2018) [32] | | Collaborative Individual Participant Data of Going-to-sleep and Stillbirth (CRIBSS) analysis | | | |
|---|---|---|---|---|---|---|---|---|---|---|---|---|---|---|
| | Case | Control | Case | Control | Case | Control | Case | Control | Case | Control | Case | Control | Univariable odds ratio (95% CI) | Adjusted odds ratio (95% CI) |
| **Total participants** | 155 (33.8) | 304 (66.2) | 103 (34.9) | 192 (65.1) | 163 (22.5) | 560 (77.5) | 288 (28.2) | 733 (71.8) | 142 (23.3) | 468 (76.7) | 851 (27.4) | 2257 (72.6) | | |
| Positive ≥10–15 | 16 (10.3) | 29 (9.5) | - | - | 25 (15.4) | 68 (12.2) | 33 (11.5) | 93 (12.7) | 24 (18.6) | 61 (16.0) | 98 (13.4) | 251 (12.7) | 1.09 (0.84–1.41) | - |
| Positive >15 | 3 (1.9) | 5 (1.6) | | | 9 (5.6) | 13 (2.3) | 9 (3.2) | 24 (3.3) | 9 (7.0) | 19 (5.0) | 30 (4.1) | 61 (3.1) | 1.41 (0.89–2.23) | - |
| **Sleep quality (last month)** | | | | | | | | | | | | | | |
| Very good | 20 (12.9) | 32 (10.5) | - | - | 25 (15.3) | 92 (16.4) | 33 (11.5) | 49 (6.7) | 9 (6.7) | 18 (4.5) | 87 (11.7) | 191 (9.6) | 1.26 (0.94–1.67) | - |
| Good to average | 79 (51.0) | 170 (55.9) | - | - | 98 (60.1) | 362 (64.6) | 146 (50.7) | 377 (51.5) | 83 (61.5) | 190 (47.9) | 406 (54.8) | 1099 (55.1) | 1 | - |
| Fairly bad | 42 (27.1) | 79 (26.0) | - | - | 27 (16.6) | 80 (14.3) | 82 (28.5) | 220 (30.1) | 33 (24.4) | 159 (40.1) | 184 (24.8) | 538 (27.0) | 0.93 (0.75–1.14) | - |
| Very bad | 14 (9.0) | 23 (7.6) | - | - | 13 (8.0) | 26 (4.7) | 27 (9.4) | 86 (11.8) | 10 (7.4) | 30 (7.6) | 64 (8.7) | 165 (8.3) | 1.03 (0.75–1.42) | - |
| **Frequency of getting up to use the toilet overnight (last month)** | | | | | | | | | | | | | | |
| <1 | 19 (12.3) | 36 (11.8) | - | - | 12 (3.4) | 39 (7.0) | 38 (13.2) | 78 (10.6) | 5 (3.7) | 16 (4.1) | 74 (10.0) | 169 (8.5) | 1.07 (0.80–1.45) | - |
| ≥1 | 136 (87.7) | 268 (88.2) | - | - | 151 (92.6) | 521 (93.0) | 250 (86.8) | 655 (89.4) | 130 (96.3) | 376 (95.9) | 667 (90.0) | 1820 (91.5) | 1 | - |

Data are number (percentage) or median (IQR). TASS = The Auckland Stillbirth Study. SSS = Sydney Stillbirth Study. MCSS = New Zealand Multicentre Stillbirth Study. MiNESS = Midlands and North of England Stillbirth Study. STARS = Study of Trends and Associated Risks for Stillbirth Study. Participants with missing data were excluded from the multivariable models. No imputation for missing data. Multivariable models are adjusted for matching terms (gestation at interview or survey in controls, and diagnosis of stillbirth for cases), study and site, age, BMI, ethnicity, parity, education, marital status, pre-existing hypertension or diabetes, smoking, drug use, baby birthweight centile, fetal movement, supine going-to-sleep position, habitual snoring, the Berlin Questionnaire, restless sleep, sleep duration, and daytime naps.

A positive BQ was independently associated with late stillbirth (Table 2), although this association was no longer significant when BMI >30 was excluded from the BQ (Model 2). This aligns with the suggestion [48, 49] that the BQ used in pregnant women is a proxy for BMI during late pregnancy, due to BMI being a component of the BQ. Indeed, the BQ performs poorly as a screening tool for objective SDB measures during pregnancy, with a 2018 meta-analysis [47] of six studies (n = 604 participants) reporting poor to fair BQ performance during pregnancy with an overall probability of OSA occurrence of 38% if a pregnant woman has a positive BQ. This range may be due to the BQ including risk factors that do not apply to pregnant women (male gender, age >50 years) and because weight gain is relevant for all pregnancies. Furthermore, symptoms of SDB progress with gestation, and there are differing opinions about the optimal timing of the BQ during pregnancy [47].

Long sleep duration >9 hours was also associated with late stillbirth (Table 2), and this association has previously been reported in two case-control studies [26, 32]. While the reason is uncertain, it is plausible that prolonged periods of aortocaval compression [36, 37] during maternal sleep may be a factor. It is also possible that an unmeasured confounder associated with long third trimester sleep (e.g. working night shifts or no paid employment) [50] may lengthen the duration of maternal sleep over the last month and contribute to stillbirth. The

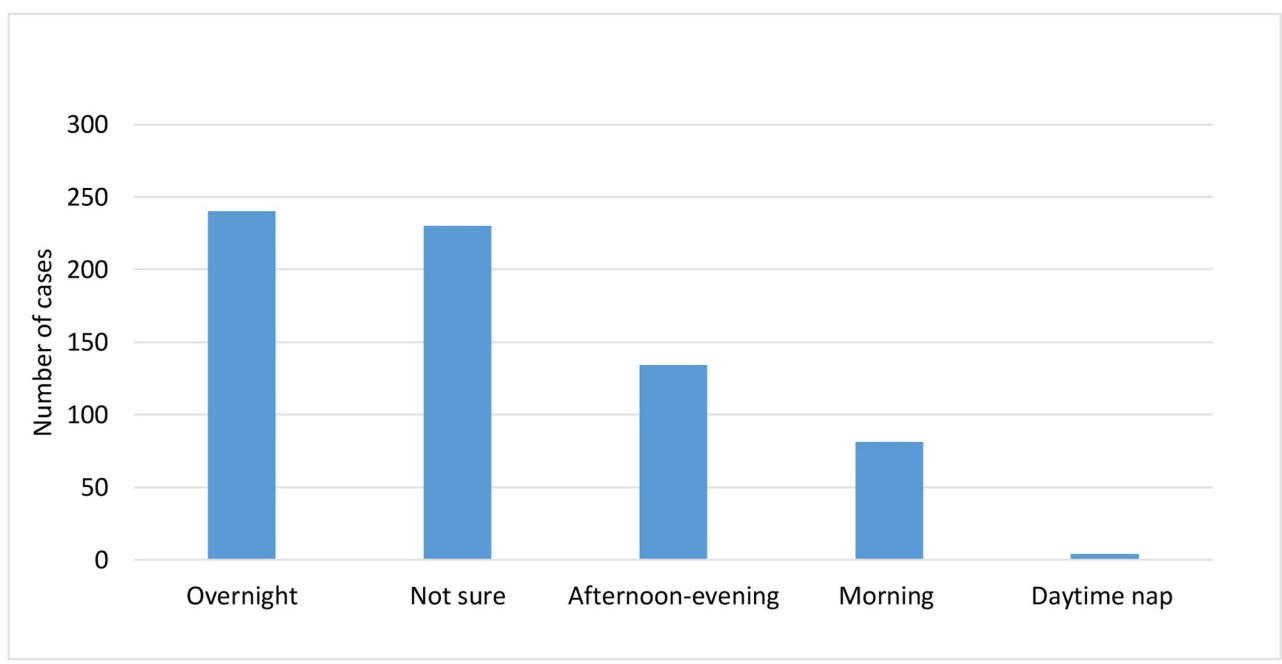

**Fig 2. Women who had a stillbirth and their perception of timing of the death.** Data are n = 689.

definition of long duration in the individual case-control studies is also inconsistent, ranging from >8 hours [26] to >9 hours [32]. This range may be due to lack of consensus about what is considered normal sleep duration in healthy pregnancy [51], although self-reported time to sleep in the third trimester is similar to objectively measured sleep duration [51] and maternal

**Table 3. Analysis for interaction between supine going-to-sleep position, and habitual snoring, the Berlin Questionnaire, sleep duration >9 hours and restless sleep greater than average.**

| | Sleep factor | Supine position | n | % | Univariable odds ratio (95% CI) | Univariable interaction p value | Multivariable odds ratio (95% CI) | Multivariable interaction p value |
|---|---|---|---|---|---|---|---|---|
| **Habitual snoring** | Yes | Yes | 17 | 0.8 | 1.44 (0.50–4.18) | **0.04** | 1.03 (0.29–3.64) | **0.001** |
| | Yes | No | 382 | 18.4 | 1.49 (1.16–1.92) | | 1.17 (0.82–1.66) | |
| | No | Yes | 75 | 3.6 | 3.37 (2.08–5.45) | | 3.75 (2.02–6.95) | |
| | No | No | 1606 | 77.2 | 1 | | 1 | |
| **Positive Berlin Questionnaire** | Yes | Yes | 22 | 1.0 | 3.44 (1.45–8.12) | 0.56 | - | - |
| | Yes | No | 510 | 23.1 | 1.56 (1.24–1.95) | | - | |
| | No | Yes | 75 | 3.4 | 2.96 (1.83–4.80) | | - | |
| | No | No | 1599 | 72.5 | 1 | | - | |
| **Sleep duration >9hrs** | Yes | Yes | 12 | 0.5 | 4.10 (1.28–13.13) | 0.99 | - | - |
| | Yes | No | 178 | 6.9 | 1.55 (1.11–2.15) | | - | |
| | No | Yes | 114 | 4.4 | 2.63 (1.78–3.88) | | - | |
| | No | No | 2271 | 88.2 | 1 | | - | |
| **Restless sleep greater than average** | Yes | Yes | 34 | 1.5 | 1.19 (0.56–2.54) | 0.10 | - | - |
| | Yes | No | 916 | 41.7 | 3.45 (2.05–5.80) | | - | |
| | No | Yes | 63 | 2.9 | 0.75 (0.61–0.93) | | - | |
| | No | No | 1186 | 53.9 | 1 | | - | |

Participants with missing data were excluded from the analysis. No imputation for missing data.

estimates of sleep duration increases in accuracy with increasing duration of sleep [52]. There was no association between short sleep duration during last month and late stillbirth, despite an independent association with short sleep on the night before stillbirth in three case-control studies [26, 30, 31]. This discrepancy may be due to a potentially fatal fetal event (e.g. pre-labour contractions for an acutely compromised fetus) that may shorten sleep on the night before stillbirth [53].

Daily daytime naps were also associated with a 1.5-fold increase in the odds of late stillbirth compared with no daytime naps (Table 2), and this finding is consistent with individual studies [26, 30, 31]. The physiology behind this is unknown and cannot be explained by overnight sleep duration or daytime sleepiness, as daily naps remained significant when we controlled for these factors. However, we speculate that daily naps in late pregnancy may increase the duration of maternal inactivity, potentially increasing the amount of time that the women spend in the supine position and therefore the duration of aortocaval compression, which when combined with the blood pressure dips that occur during third trimester sleep [54], may further compromise a vulnerable fetus [34].

Our finding of a 38% reduction in the odds of late stillbirth for women who reported restless sleep more than average during the last month is novel (Table 2). We speculate that this may be due to maternal body movement facilitating maternal-fetal blood flow, potentially abating adverse fetal effects of aortocaval compression [37, 55]. Furthermore, while maternal hypotension is known to have adverse fetal consequences, such as lower birth weight and stillbirth [56–59], increased third trimester arousals related to snoring [60] may assuage prolonged periods of relative hypotension, as deep sleep is commensurate with the lowest overnight blood pressure and arousal with increased blood pressure [61]. Our finding of a protective association between restless sleep more than average and late stillbirth aligns with an international case-control study [32] that reported non-restless sleep in the last month was associated with a 1.7-fold increase in odds of late stillbirth. Similarly, getting up to use the toilet on the night before stillbirth is associated with a 2-fold reduction in late stillbirth [26, 30, 31], suggesting that maternal body movement on the night before stillbirth may mitigate the effects of a hypoxic event on the fetus [62].

Certainly, pregnant women are susceptible to the development of sleep disturbances, commonly reduced quality and duration of sleep, night waking, daytime sleepiness, and snoring [45, 63]. Causes are most likely to be hormonal and physiological changes of pregnancy, including increased oxygen consumption and metabolic rate, lower overall oxygen reserve, nasopharyngeal oedema, vasomotor rhinitis, and weight gain, which contribute to narrowing of upper airway, reduced functional residual capacity due to diaphragmatic pressure by the growing fetus, and increased arousals during sleep [60, 63]. These physiological changes are exacerbated as pregnancy progresses and when combined with obesity, advanced maternal age, and supine sleep position [64–66].

Conversely, late pregnancy may provide some protection from SDB, with increased respiratory drive [67], alteration in the cyclical sleep pattern with decreased rapid eye movement (REM) sleep [60, 63, 68], and preference for a lateral sleep position [26, 30, 45, 69]. These may be factors contributing to our finding of a significant interaction between habitual snoring during the last month and supine going-to-sleep position, with a lower odds of late stillbirth than expected in women who reported both during the last month. While this may be a chance finding due to low prevalence, with 17 (12 controls and 5 cases) of 92 women reporting habitual snoring and a supine going-to-sleep position, this could also be explained by the women being woken by a sleep companion or experiencing a self-arousal due to snoring, and moving from the supine to a lateral position, which is known to reduce third trimester snoring in obese women [23] and late stillbirth risk [24].

## Strengths and limitations

A limitation of the IPD meta-analysis is that not all participating studies had data for all sleep measures. Minor differences in the design of the individual studies also limited the inclusion of some covariates. Our search had no language restriction and an eligible study from India was identified, however, there was no response from authors or journal editors to repeated invitations to participate. No other eligible randomised trials, prospective cohort studies or studies from low-income countries were identified, thus participating studies were all case-control studies from high-income countries. A limitation of case-control studies include the retrospective data collection which is subject to potential recall bias, although as the relationship between late stillbirth and maternal sleep is not universally well known by pregnant women, systematic bias is unlikely. The longer length of time before interview for cases may have influenced their recall compared to controls, however, case recall is unlikely to be biased towards an association with SDB, with self-reports from a single night of sleep having similar bias and calibration as 'usual' sleep [70]. Use of self-reported symptoms of SDB, rather than objective measures using polysomnography may also be considered a limitation. However, self-report of snoring is strongly and reliably associated with the severity of OSA obtained from polysomnography in non-pregnant [71] and pregnant women [46], therefore self-report is useful for large scale studies where routine access to polysomnography in late pregnancy is costly and impractical.

## Conclusion

This IPD meta-analysis adds to the evidence on maternal sleep and late stillbirth, using the best available data on the association of SDB and maternal sleep patterns with the risk of late stillbirth. These findings demonstrate that self-reported maternal snoring, a positive BQ screen excluding BMI, daytime sleepiness, sleep quality, and getting up to use the toilet, are not independently associated with late stillbirth last month. Long sleep duration >9 hours and daily daytime naps are independent risk factors, while sleep more restless than average may reduce the odds of late stillbirth. There is an urgent need to better understand factors associated with long sleep duration and daily daytime naps before recommendations can be made to pregnant women. Meanwhile, pregnant women may be reassured that the commonly reported increased restlessness of sleep during late pregnancy may be physiological and is associated with a reduced risk of late stillbirth.

## Supporting information

**S1 Fig. Chart of available data from contributing studies.**
(DOCX)

**S1 Table. Participant level characteristics and non-sleep late stillbirth risk factors in participating case-control studies and pooled IPD meta-analysis.**
(DOCX)

**S1 Checklist. PRISMA-P (Preferred reporting items for systematic review and meta-analysis protocols) 2015 checklist: Recommended items to address in a systematic review protocol\*.**
(DOC)

**S1 Protocol.**
(PDF)

## Acknowledgments

We would like to thank the women who participated in the individual studies and the research midwives who conducted the interviews.

## Author Contributions

**Conceptualization:** Robin S. Cronin, Jessica Wilson, Adrienne Gordon, Minglan Li, Vicki M. Culling, Camille H. Raynes-Greenow, Alexander E. P. Heazell, Tomasina Stacey, Lisa M. Askie, Edwin A. Mitchell, John M. D. Thompson, Lesley M. E. McCowan, Louise M. O'Brien.

**Data curation:** Robin S. Cronin, Jessica Wilson, Adrienne Gordon, Minglan Li, Camille H. Raynes-Greenow, Alexander E. P. Heazell, Tomasina Stacey, Edwin A. Mitchell, John M. D. Thompson, Lesley M. E. McCowan, Louise M. O'Brien.

**Formal analysis:** Robin S. Cronin, Jessica Wilson, John M. D. Thompson.

**Funding acquisition:** Robin S. Cronin, Adrienne Gordon, Minglan Li, Vicki M. Culling, Camille H. Raynes-Greenow, Alexander E. P. Heazell, Tomasina Stacey, Edwin A. Mitchell, Lesley M. E. McCowan.

**Investigation:** Robin S. Cronin, Jessica Wilson, Adrienne Gordon, Minglan Li, Camille H. Raynes-Greenow, Alexander E. P. Heazell, Tomasina Stacey, Edwin A. Mitchell, John M. D. Thompson, Lesley M. E. McCowan, Louise M. O'Brien.

**Methodology:** Robin S. Cronin, Jessica Wilson, Adrienne Gordon, Minglan Li, Camille H. Raynes-Greenow, Alexander E. P. Heazell, Tomasina Stacey, Lisa M. Askie, Edwin A. Mitchell, John M. D. Thompson, Lesley M. E. McCowan, Louise M. O'Brien.

**Project administration:** Robin S. Cronin, Jessica Wilson, Adrienne Gordon, Minglan Li, Alexander E. P. Heazell, Tomasina Stacey, Edwin A. Mitchell, John M. D. Thompson, Lesley M. E. McCowan.

**Supervision:** John M. D. Thompson, Lesley M. E. McCowan, Louise M. O'Brien.

**Writing – original draft:** Robin S. Cronin.

**Writing – review & editing:** Robin S. Cronin, Jessica Wilson, Adrienne Gordon, Minglan Li, Vicki M. Culling, Camille H. Raynes-Greenow, Alexander E. P. Heazell, Tomasina Stacey, Lisa M. Askie, Edwin A. Mitchell, John M. D. Thompson, Lesley M. E. McCowan, Louise M. O'Brien.

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
