## [Decision Letter · Decision Letter 0]

5 Jan 2020

PONE-D-19-22011

Associations between symptoms of sleep-disordered breathing and maternal sleep patterns with late stillbirth: findings from an individual participant data meta-analysis.

PLOS ONE

Dear Mrs Cronin,

Thank you for submitting your manuscript to PLOS ONE. After careful consideration, we feel that it has merit but does not fully meet PLOS ONE’s publication criteria as it currently stands. Therefore, we invite you to submit a revised version of the manuscript that addresses the points raised during the review process.

We would appreciate receiving your revised manuscript by Feb 19 2020 11:59PM. To enhance the reproducibility of your results, we recommend that if applicable you deposit your laboratory protocols in protocols.io, where a protocol can be assigned its own identifier (DOI) such that it can be cited independently in the future. For instructions see: http://journals.plos.org/plosone/s/submission-guidelines#loc-laboratory-protocols

We look forward to receiving your revised manuscript.

Kind regards,

Claudio Liguori

Academic Editor

PLOS ONE

Journal Requirements:

Mrs Cronin reports grants from Health Research Council of New Zealand (12/372); Cure Kids (5357); Mercia Barnes Trust; Nurture Foundation; University of Auckland Faculty Research Development Fund (3700696)., grants from 2016 TransTasman Red Nose/Curekids (6601), grants from Sir John Logan Campbell Medical Trust, during the conduct of the study;

Ms. Wilson reports grants from 2016 TransTasman Red Nose/Curekids (6601);

Dr. Gordon reports grants from Stillbirth Foundation Australia, during the conduct of the study; other from NHMRC Early Career Fellowship #1089898,  outside the submitted work;

Dr. Li reports grants from Health Research Council of New Zealand (12/372); Cure Kids (5357); Mercia Barnes Trust; Nurture Foundation; University of Auckland Faculty Research Development Fund (3700696)., grants from 2016 TransTasman Red Nose/Curekids (6601), during the conduct of the study;

Dr. Culling has nothing to disclose;

Associate Professor Raynes-Greenow reports grants from Stillbirth Foundation Australia, during the conduct of the study; other from NHMRC Career Development Fellowship #1087062, outside the submitted work;

Professor Heazell reports grants from Action Medical Research, during the conduct of the study; grants from Tommy's, grants from NIHR, outside the submitted work;

Dr. Stacey reports grants from Cure Kids (3537), Nurture Foundation, Auckland District Health Board Charitable Trust., grants from Health Research Council of New Zealand (12/372); Cure Kids (5357); Mercia Barnes Trust; Nurture Foundation; University of Auckland Faculty Research Development Fund (3700696)., during the conduct of the study;

Professor Askie has nothing to disclose;

Professor Mitchell reports grants from Cure Kids (Grant 3537), Nurture Foundation, Auckland District Health Board Trust., grants from Health Research Council of New Zealand (12/372); Cure Kids (5357); Mercia Barnes Trust; Nurture Foundation; University of Auckland Faculty Research Development Fund (3700696)., grants from 2016 TransTasman Red Nose/Curekids (6601), during the conduct of the study; other from Cure Kids, outside the submitted work;

Associate Professor Thompson reports grants from Cure Kids (3537), Nurture Foundation, Auckland District Health Board Charitable Trust., grants from Health Research Council of New Zealand (12/372); Cure Kids (5357); Mercia Barnes Trust; Nurture Foundation; University of Auckland Faculty Research Development Fund (3700696)., grants from 2016 TransTasman Red Nose/Curekids (6601), during the conduct of the study;

Professor McCowan reports grants from Cure Kids (537), Nurture Foundation, Auckland District Health Board Trust., grants from Health Research Council of New Zealand (12/372); Cure Kids (5357);  Mercia Barnes Trust; Nurture Foundation; University of Auckland Faculty Research Development Fund (3700696)., grants from 2016 TransTasman Red Nose/Curekids (6601), during the conduct of the study.

Associate Professor O'Brien reports grants from American Sleep Medicine Foundation, grants from ResMed, outside the submitted work;

We note that you received funding from a commercial source: ResMed

Additional Editor Comments:

Only some minor changes have been requested by the Reviewers; please modify the manuscript as requested.

Reviewers' comments:

Reviewer's Responses to Questions

**Comments to the Author**

1. Is the manuscript technically sound, and do the data support the conclusions?

Reviewer #1: Yes

Reviewer #2: Yes

2. Has the statistical analysis been performed appropriately and rigorously? 

Reviewer #1: Yes

Reviewer #2: Yes

3. Have the authors made all data underlying the findings in their manuscript fully available?

Reviewer #1: No

Reviewer #2: Yes

4. Is the manuscript presented in an intelligible fashion and written in standard English?

Reviewer #1: Yes

Reviewer #2: Yes

5. Review Comments to the Author

Reviewer #1: The aim of this work is to find the relationship of symptom of SDB and late stillbirth from previously done case-control studies that is available and pool them together using statistical technique. The point that this paper is a successful collaboration between large teams around the globe combining with a robust individual patient-level data analysis technique is interesting. I congratulate the authors for their efforts.

Overall I find the manuscript to be of good quality, however some minor revision points may need to be done:

1. Study selection process report need to be more transparent, i.e. the authors should describe number of study that were excluded (fig. 1 may need to be revised) how many duplicates were there?, why 124 studies did not meet the eligibility criteria? There should be more detail on the number and reason of studies that were excluded, to provided that the author has established all available evidences to support the meta-analysis. Also, for one study that is not participating, is there any difference in characteristic of that particular study compared to participating studies? (The author may need to describe about this in the result)

2. Case control studies were prone to recall bias and combining case-control in meta-analysis can still be susceptible to this. I wonder if there any other supporting evidences from cohort study? If there is not the author should specifically mention about the result of the assessment of quality of the evidence in the result, and emphasis on the possibility of bias upon making conclusion, which may need a large pregnancy cohort to be done.

Again, I sincerely admire the authors' efforts into the collaborating process.

Reviewer #2: Comments to the Author

-How the duration of overnight sleep was evaluated? Did you use a standardized sleep log?

-How did you evaluated the sleep position (supine and not supine)? Is this data only self-reported? Or did you use some instruments to evaluate it? If only self-reported, it can probably be inaccurate because you can’t know the real position throughout the night. Can you specify it?

-You suggest that a reduction in the odds of late stillbirth for women who reported restless sleep during the last month may be is due to maternal body movement facilitating maternal-fetal blood flow, potentially abating adverse fetal effects of aortocaval compression. Considering that the supine position is associated with an increase in aortocaval compression, have you evaluated interactions between restless sleep and sleep position to understand if the reduced risk of late stillbirth in these cases can be related to the position?

-You report that the combined effect of supine going-to-sleep position and habitual snoring resulted in a reduced odds of late stillbirth in the multivariable model and you explain this result assuming that snoring can cause arousal that can cause a change of position from supine to lateral. It would be interesting to test this hypothesis with an objective assessment of the position.

-You report that getting up to use the toilet in the night is associated with a reduced risk of late stillbirth, suggesting that maternal body movement on the night may mitigate the effects of a hypoxic event on the fetus. Considering that getting up to use the toilet in the night can be also a clinical manifestation of SBD, have you analyzed interactions between the use of the toilet in the night and Berlin Questionnaire (excluding BMI)?

-Non using objective measures (polysomnography) for SBD’s evaluation may be a limitation as you specified. Didn’t you evaluate the possibility to assess the polysomography, if not possibile in all the women, in some of them such as patients with positive Berlin Questionnaire (excluding BMI), in patients with restless sleep and in patients getting up to use the toilet?

6. PLOS authors have the option to publish the peer review history of their article (what does this mean?). If published, this will include your full peer review and any attached files.

Reviewer #1: No

Reviewer #2: Yes: Dott.ssa Francesca Furia

---

## [Author Response · Author response to Decision Letter 0]

15 Feb 2020

The authors thank the editorial team and reviewers’ for their comments. The manuscript has been revised according to the comments.

EDITORIAL FEEDBACK

- AUTHOR RESPONSE: We agree, and have addressed these additional requirements.

2) Please include captions for your Supporting Information files at the end of your manuscript, and update any in-text citations to match accordingly. Please see our Supporting Information guidelines for more information: http://journals.plos.org/plosone/s/supporting-information

- AUTHOR RESPONSE: We agree, and have added detail to the captions for the Supporting Information files at the end of our manuscript.

3) Thank you for stating the following in the Financial Disclosure section… We note that you received funding from a commercial source: ResMed. Please provide an amended Competing Interests Statement that explicitly states this commercial funder, along with any other relevant declarations relating to employment, consultancy, patents, products in development, marketed products, etc. Within this Competing Interests Statement, please confirm that this does not alter your adherence to all PLOS ONE policies on sharing data and materials by including the following statement: "This does not alter our adherence to PLOS ONE policies on sharing data and materials.” (as detailed online in our guide for authors http://journals.plos.org/plosone/s/competing-interests). If there are restrictions on sharing of data and/or materials, please state these. Please note that we cannot proceed with consideration of your article until this information has been declared. Please include your amended Competing Interests Statement within your cover letter. We will change the online submission form on your behalf.

- AUTHOR RESPONSE: We agree, and have amended our Competing Interests Statement as follows:

“Associate Professor O'Brien reports grants from ResMed outside the submitted work. This does not alter our adherence to PLOS ONE policies on sharing data and materials.” 

This amended Competing Interests Statement has been included within our cover letter. Thank you for changing the online submission form on our behalf.

4) We note that you have indicated that data from this study are available upon request. PLOS only allows data to be available upon request if there are legal or ethical restrictions on sharing data publicly. For information on unacceptable data access restrictions, please see http://journals.plos.org/plosone/s/data-availability#loc-unacceptable-data-access-restrictions.

- AUTHOR RESPONSE: We have provided additional details regarding the ethical restrictions on sharing our data publicly in our revised cover letter as follows:

“Data cannot be shared publicly beyond the Collaborative Individual Participant Data Meta-analysis of Sleep and Stillbirth (CRIBSS) group as no individual participating study obtained consent from participants to make the data publically available. Furthermore, because stillbirth is uncommon there is potential for participants to be identifiable. Contact information for the CRIBSS Data Access Committee is The CRIBSS Data Centre, Department of Obstetrics and Gynaecology, Faculty of Medical and Health Sciences, University of Auckland, Private Bag 92019, Auckland Mail Centre, Auckland 1142.”

Thank you for updating our Data Availability statement.

REVIEWER FEEDBACK

REVIEWER # 1

1.1) The aim of this work is to find the relationship of symptom of SDB and late stillbirth from previously done case-control studies that is available and pool them together using statistical technique. The point that this paper is a successful collaboration between large teams around the globe combining with a robust individual patient-level data analysis technique is interesting. I congratulate the authors for their efforts. Overall I find the manuscript to be of good quality, however some minor revision points may need to be done:

- AUTHOR RESPONSE: Thank you.

1.2) Study selection process report need to be more transparent, i.e. the authors should describe number of study that were excluded (fig. 1 may need to be revised) how many duplicates were there?, why 124 studies did not meet the eligibility criteria? There should be more detail on the number and reason of studies that were excluded, to provided that the author has established all available evidences to support the meta-analysis. Also, for one study that is not participating, is there any difference in characteristic of that particular study compared to participating studies? (The author may need to describe about this in the result)

- AUTHOR RESPONSE: We agree, and have revised “Figure 1 PRISMA study population flow chart,” which now includes the number of duplicate articles, and participant level inclusion and exclusion criteria. We have also added more detail to the Strengths and Limitations section as follows:

“Our search had no language restriction and an eligible study from India was identified, however, there was no response from authors or journal editors to repeated invitations to participate. No other eligible randomised trials, prospective cohort studies or studies from low-income countries were identified, thus participating studies were all case-control studies from high-income countries.”

Please note that a December 23, 2019 update for Beall's List of Predatory Journals and Publishers states that the journal that published the non-participating study from India (the International Journal of Reproduction, Contraception, Obstetrics and Gynecology) has been included with “journals that were not originally on the Beall's list but may be predatory.” https://beallslist.net/standalone-journals/

1.3) Case control studies were prone to recall bias and combining case-control in meta-analysis can still be susceptible to this. I wonder if there any other supporting evidences from cohort study? If there is not the author should specifically mention about the result of the assessment of quality of the evidence in the result, and emphasis on the possibility of bias upon making conclusion, which may need a large pregnancy cohort to be done.

- AUTHOR RESPONSE: We agree that case-control studies have the limitation of recall bias, and that supporting evidence from cohort studies should be reported where available. This has been explained in the Strengths and Limitations section, and we have clarified that that it is the relationship between late stillbirth and maternal sleep that is not universally well known as follows:

“A limitation of case-control studies include the retrospective data collection which is subject to potential recall bias, although as the relationship between late stillbirth and maternal sleep is not universally well known by pregnant women, systematic bias is unlikely. The longer length of time before interview for cases may have influenced their recall compared to controls, however, case recall is unlikely to be biased towards an association with SDB, with self-reports from a single night of sleep having similar bias and calibration as ‘usual’ sleep [70]. Use of self-reported symptoms of SDB, rather than objective measures using polysomnography may also be considered a limitation. However, self-report of snoring is strongly and reliably associated with the severity of OSA obtained from polysomnography in non-pregnant [71] and pregnant women [46]; therefore self-report is useful for large-scale studies where routine access to polysomnography in late pregnancy is costly and impractical.”

We also agree that the assessment of the quality of the evidence should be reported. Under the Materials and Methods section, we have already provided a reference to the risk of bias tool and risk of bias report used for this study as follows:

“This IPD meta-analysis was registered with the PROSPERO register of systematic reviews (CRD42017047703) and followed the IPD meta-analysis protocol [38], search strategy [24], risk of bias for non-randomised studies (ROBINS-E) tool [39], and published results [24].”

REVIEWER #2

2.1) How the duration of overnight sleep was evaluated? Did you use a standardized sleep log?

- AUTHOR RESPONSE: Sleep data, including sleep duration, were collected by self-report in all participating studies. No study collected sleep data via standardised sleep log or actigraphy. To make this clear, we have added the words “self-report” to the Materials and Methods section as follows:

“Maternal sleep data were collected by self-report via face-to-face interview [26, 27, 30, 31] or online survey [32] within six weeks after stillbirth in cases or at a matched gestation in controls”

2.2) a) How did you evaluated the sleep position (supine and not supine)? Is this data only self-reported? Or did you use some instruments to evaluate it? b) If only self-reported, it can probably be inaccurate because you can’t know the real position throughout the night. Can you specify it.

- AUTHOR RESPONSE:

a) Please see answer to 2.1.

b) The participating studies were all questionnaire based, data on maternal sleep position throughout the night was not collected. The supine/non-supine going-to-sleep position variable in our manuscript refers only to going-to-sleep position. The reason that going-to-sleep position was the chosen variable was because this is most easily recalled and modifiable for the majority of women in late pregnancy (Cronin et al., 2017, https://doi.org/10.1186/s12884-017-1378-5). We were not able to validate maternal self-report of going-to-sleep position in the individual studies, however, in some of the participating studies the investigators reported that participants often described reference points to remember their going-to-sleep position, such as facing the door. In addition, McIntyre et al’s 2016 overnight sleep study of 30 healthy women in late pregnancy (https://doi.org/10.1186/s12884-016-0905-0), reported good agreement for going-to-sleep (sleep onset) position between infared digital video and self-completed questionnaires for participants (who were not given any information about sleep position), with the majority accurately recalling their going-to-sleep position.

2.3) You suggest that a reduction in the odds of late stillbirth for women who reported restless sleep during the last month may be is due to maternal body movement facilitating maternal-fetal blood flow, potentially abating adverse fetal effects of aortocaval compression. Considering that the supine position is associated with an increase in aortocaval compression, have you evaluated interactions between restless sleep and sleep position to understand if the reduced risk of late stillbirth in these cases can be related to the position?

- AUTHOR RESPONSE: We agree that the relationship between restless sleep and supine going-to-sleep position is interesting. Therefore, we have added the analysis of interaction between supine going-to-sleep position and restless sleep greater than average last month to “Table 3: Analysis for interaction between supine going-to-sleep position, and habitual snoring, the Berlin Questionnaire, sleep duration >9 hours and restless sleep greater than average.” The interaction between supine going-to-sleep position and restless sleep was not statistically significant (p=0.98).

2.4) You report that the combined effect of supine going-to-sleep position and habitual snoring resulted in a reduced odds of late stillbirth in the multivariable model and you explain this result assuming that snoring can cause arousal that can cause a change of position from supine to lateral. It would be interesting to test this hypothesis with an objective assessment of the position.

- AUTHOR RESPONSE: We agree that an objective assessment of sleep position in women with habitual snoring during late pregnancy would be of great interest and could be evaluated in future studies. 

2.5) You report that getting up to use the toilet in the night is associated with a reduced risk of late stillbirth, suggesting that maternal body movement on the night may mitigate the effects of a hypoxic event on the fetus. Considering that getting up to use the toilet in the night can be also a clinical manifestation of SBD, have you analyzed interactions between the use of the toilet in the night and Berlin Questionnaire (excluding BMI)?

- AUTHOR RESPONSE: We agree that we reported that getting up to use the toilet on the night before stillbirth was associated with halving of the odds of late stillbirth in three individual studies. However, there was no association found between getting up to use the toilet during the month before stillbirth in the analysis reported in our manuscript.

2.6) Non using objective measures (polysomnography) for SBD’s evaluation may be a limitation as you specified. Didn’t you evaluate the possibility to assess the polysomography, if not possibile in all the women, in some of them such as patients with positive Berlin Questionnaire (excluding BMI), in patients with restless sleep and in patients getting up to use the toilet? 

- AUTHOR RESPONSE: The participating studies were all questionnaire based and any referral for polysomnography was not captured.

---

## [Decision Letter · Decision Letter 1]

11 Mar 2020

Associations between symptoms of sleep-disordered breathing and maternal sleep patterns with late stillbirth: findings from an individual participant data meta-analysis.

PONE-D-19-22011R1

Dear Dr. Cronin,

We are pleased to inform you that your manuscript has been judged scientifically suitable for publication and will be formally accepted for publication once it complies with all outstanding technical requirements.

With kind regards,

Claudio Liguori

Academic Editor

PLOS ONE

Additional Editor Comments (optional):

Reviewers' comments:

Reviewer's Responses to Questions

**Comments to the Author**

1. If the authors have adequately addressed your comments raised in a previous round of review and you feel that this manuscript is now acceptable for publication, you may indicate that here to bypass the “Comments to the Author” section, enter your conflict of interest statement in the “Confidential to Editor” section, and submit your "Accept" recommendation.

Reviewer #2: All comments have been addressed

2. Is the manuscript technically sound, and do the data support the conclusions?

Reviewer #2: Yes

3. Has the statistical analysis been performed appropriately and rigorously? 

Reviewer #2: Yes

4. Have the authors made all data underlying the findings in their manuscript fully available?

Reviewer #2: Yes

5. Is the manuscript presented in an intelligible fashion and written in standard English?

Reviewer #2: Yes

6. Review Comments to the Author

Reviewer #2: I congratulate the authors for their work. I find that the manuscript is well written, well analizyzed and of good quality.

7. PLOS authors have the option to publish the peer review history of their article (what does this mean?). If published, this will include your full peer review and any attached files.

Reviewer #2: Yes: Francesca Furia

---

## [Editor Report · Acceptance letter]

13 Mar 2020

PONE-D-19-22011R1 

Associations between symptoms of sleep-disordered breathing and maternal sleep patterns with late stillbirth: findings from an individual participant data meta-analysis. 

Dear Dr. Cronin:

I am pleased to inform you that your manuscript has been deemed suitable for publication in PLOS ONE. Congratulations! Your manuscript is now with our production department. 

With kind regards,

on behalf of

Dr. Claudio Liguori 

Academic Editor

PLOS ONE